# Mutation and Microsatellite Instability (MSI) Affect the Differential Gene Expression of Folic Acid and 5-Flourouracil Metabolism-Related Genes in Colorectal Carcinoma

**DOI:** 10.3390/curroncol32120661

**Published:** 2025-11-26

**Authors:** Muhammad Rafiqul Islam, Farzana Jasmine, Daniil Vasiljevs, Maruf Raza, Armando Almazan, Habibul Ahsan, Muhammad G. Kibriya

**Affiliations:** 1Institute for Population and Precision Health, Biological Sciences Division, University of Chicago, Chicago, IL 60637, USA; rafiq@uchicago.edu (M.R.I.); farzana@uchicago.edu (F.J.); daniil.vasiljevs@bsd.uchicago.edu (D.V.); armando.almazan@bsd.uchicago.edu (A.A.); habib@uchicago.edu (H.A.); 2Department of Pathology, Jahurul Islam Medical College, Kishoregonj 2336, Bangladesh; maruf_path@yahoo.com; 3Department of Public Health Sciences, Biological Sciences Division, University of Chicago, Chicago, IL 60637, USA

**Keywords:** colorectal carcinoma, 5-fluorouracil (5-FU), folic acid, *TYMS*, *KRAS*, *TP53*, MSI, cell cycle checkpoint

## Abstract

5-fluorouracil (5-FU) is a chemotherapy that is used in combination with other biologics to treat colorectal cancer (CRC). Folic acid increases the potency of 5-FU; however, certain patient characteristics impact the response to this treatment. This study aims to identify the characteristics that significantly affect the genes that are related to 5-FU and folic acid metabolism by utilizing data from gene expression assays on CRC samples. The significant relationships discovered in this population can help guide the development of tailored therapy for these subgroups of CRC patients.

## 1. Introduction

In 2022, an estimated 1.93 million new colorectal cancer (CRC) cases were diagnosed, and 0.94 million CRC-related deaths were reported [1]. According to the American Cancer Society (ACS), CRC is the second most common cause of cancer-related mortality in the United States, projected to account for 52,900 deaths in 2025 [2]. For patients with early CRC without evidence of distant metastasis, surgical intervention remains the primary treatment approach. Adjuvant chemotherapy using 5-fluorouracil-based (5-FU) combinations has been shown to reduce the risk of disease recurrence in stage III and high-risk stage II CRC [3]. For metastatic CRC (mCRC), treatment typically involves chemotherapy with 5-FU-based combinations alongside biologics such as bevacizumab or cetuximab and, in some cases, immunotherapy [4].

For treating CRC, 5-FU is the most widely used chemotherapeutic agent. It is a synthetic fluorinated pyrimidine (FP) analog of uracil administered yearly to over 2 million cancer patients worldwide [5,6]. Due to its structural similarity to pyrimidine, 5-FU is an antimetabolite that readily incorporates into DNA and RNA instead of thymine or uracil. This incorporation triggers cell cycle arrest during the S phase, suppresses proliferation, and activates apoptotic pathways in cancer cells [5,7]. The efficacy of 5-FU depends on multiple factors, including its molecular targets, cellular transport mechanisms, the body’s detoxification capacity, and functional DNA damage repair pathways [8,9,10,11,12]. Its effectiveness is also influenced by intact apoptotic processes, nucleotide synthesis pathways essential for DNA replication, cell adhesion properties, and epithelial–mesenchymal transition (EMT) regulators [13]. In addition, folic acid (FA) plays a critical role in enhancing the therapeutic effectiveness of 5-FU by facilitating the inhibition of thymidylate synthase, which is a key enzyme required for DNA synthesis [14]. When administered alongside 5-FU, folate is converted into active reduced folate forms, such as 5,10-methylene tetrahydrofolate, which stabilizes the ternary complex formed between thymidylate synthase and the active metabolites of 5-FU [15]. This stabilization prolongs the binding of 5-FU to the enzyme, thereby amplifying its ability to block DNA synthesis and induce cytotoxic effects in rapidly dividing cancer cells. Without folate, this complex is less stable, and the antitumor efficacy of 5-FU is markedly reduced [14]. Thus, folic acid supplementation, commonly given as leucovorin (folinic acid), is essential for maximizing the potency of 5-FU and improving treatment outcomes in CRC patients. Moreover, biological factors significantly impact 5-FU response, including patient age, gender, tumor site, histological grade, disease stage, Kirsten rat sarcoma virus (*KRAS*) mutation status, microsatellite instability (MSI), telomere length, and tumor-infiltrating lymphocytes (TILs) [4,16,17,18,19,20,21,22].

MSI is a form of genomic instability characterized by length alterations in short, tandemly repeated DNA sequences (microsatellites) due to insertion or deletion errors during DNA replication. Under normal conditions, the DNA mismatch repair (MMR) system—comprising *MLH1*, *MSH2*, *MSH6*, and *PMS2*—detects and corrects such errors. Deficiency of DNA mismatch repair (MMR-D) results in error accumulation and the MSI phenotype [23]. MSI occurs in approximately 15% of CRC cases, most often sporadically, although 2–4% are associated with Lynch syndrome, which is an autosomal dominant disorder caused by germline MMR gene mutations [24]. The prevalence of MSI is stage-dependent, being higher in early-stage CRC (≈20% in stages I–II, 12% in stage III) and lower in metastatic disease (4–5%) [25]. MSI-High (MSI-H)/MMR-D CRCs display distinctive features, including a bimodal age distribution, female predominance, proximal colon location, and *KRAS* wild-type status [26]. Moreover, MSI-H tumors are associated with a more favorable prognosis in early-stage CRC and with limited benefit from adjuvant 5-FU therapy in early-stage disease [27]. Overall, MSI is an important biomarker for CRC with significant diagnostic, prognostic, and predictive value [25].

Therefore, understanding the interactions between MSI and other variables such as age, sex, and disease stage is essential, not only for gaining deeper insight into tumor biology, but also for generating hypotheses that can inform precision medicine approaches based on molecular findings from clinical samples. In this study, we aimed to (a) determine the differential expression of FA and 5-FU-related genes in CRC; (b) assess whether the differential expression of gene pathways related to 5-FU and FA metabolism is influenced by different molecular features of the tumor, including MSI and mutational status and the key clinical and pathological variables; and (c) explore whether such molecular profiling can enhance our understanding of CRC pathogenesis and provide a molecular basis for 5-FU resistance, ultimately supporting more informed selection of patients for tailored therapeutic strategies.

## 2. Materials and Methods

The tissue samples from 71 CRC patients (male = 43 and female = 28) used in this study were collected from the Department of Pathology, Bangabandhu Sheikh Mujib Medical University (BSMMU), Dhaka, Bangladesh, at different times, spanning from December 2009 to May 2016. All consecutive patients who were referred to for surgical intervention for sporadic CRC during each collection period were included. None of them knew about distal metastasis. None of them received any chemotherapy and/or radiotherapy prior to surgery. There were no other exclusion criteria. These patients were included in our previous studies [28,29,30,31]. A surgical pathology fellow collected all the surgically resected specimens directly from the operating room. For each patient, fresh specimens were collected from resected tumors (referred to as CRC in this article) and surrounding normal-appearing colon tissue (referred to as non-lesional), 5–10 cm from the tumor. For each tissue sample, one part was preserved as fresh frozen, and the other part was preserved in RNA*later* stabilization solution (Invitrogen by Fisher Scientific, Waltham, MA, USA) and kept frozen for gene expression study. Samples were shipped on dry ice to the University of Chicago and stored at −80 °C until extraction. We collected samples from a total of 165 CRC patients. For the first 71 consecutive patients, we generated gene expression data using microarray platforms in the past. We have included the first 71 CRC patients in this study. Patient characteristics are shown in Appendix A. RNA was extracted from tissue preserved in RNA*later* using the RiboPure kit (Qiagen, Germantown, MD, USA) following the manufacturer’s recommended protocol. DNA was extracted from the fresh frozen tissue using the Puregene Core kit (Qiagen, Germantown, MD, USA).

### 2.1. Gene Selection and Functional Relevance

To investigate the 5-FU response in CRC, we selected a comprehensive panel of genes spanning metabolism, apoptosis, cell cycle, DNA repair, drug transport, metastatic plasticity, stress adaptation, metabolic reprogramming, signaling cascades, and tumor suppression (for a complete list of the 180 gene names, gene symbols, and categories, please see the Appendix A). Genes implicated in 5-FU metabolism (*DPYD*, *TYMS*, *UCK2*, *UMPS*, *UPP1*) were prioritized due to their pivotal role in regulating drug activation, degradation, and therapeutic efficacy, as dysregulation directly influences sensitivity and resistance [32]. In parallel, apoptotic regulators (*AKT1*, *APAF1*, *BAX*, *BBC3*, *BCL2* family members, *BID*, *BIRC5*, *CASP3*, *CASP7*, *CASP8*, *CASP9*, *CYCS*, *DAPK1*, *DFFA*, *FAS*/*FASLG*, *GADD45A*, *MAPK1*, *MCL1*, *NOXA1*, *PIK3CA*, *RASSF1*, *TNFRSF10A*, *TNFRSF10B*, *TNFSF10*, *TP53*) were included, as they govern intrinsic and extrinsic apoptotic pathways whose imbalance promotes tumor cell survival and chemoresistance [33]. Their imbalance allows tumor cells to evade programmed cell death and contributes to chemo-resistance [34]. Similarly, cell cycle checkpoint regulators (*ATR*, *AURKA*, *AURKB*, *BUB1*, *BUB3*, *CCNB1*, *CCND1*, *CDC2*, *CDC20*, *CDC25C*, *CDK4*, *CDK6*, *CDKN1A*, *CDKN2A*, *CHEK1*, *CHEK2*, *E2F1*, *MAD2L1*, *PLK1*, *RB1*, *TTK*, *WEE1*) were selected as they safeguard G1/S, G2/M, and spindle assembly transitions, with dysregulation driving unchecked proliferation [35,36]. DNA repair genes (*ATM*, *BRCA1*, *BRCA2*, *BRIP1*, *ERCC1*, *ERCC4*, *ERCC5*, *MLH1*, *MSH2*, *MSH6*, *PARP1*, *PARP2*, *PMS2*, *RAD50*, *RAD51*) were incorporated due to their essential role in mismatch repair, homologous recombination, and nucleotide excision repair; mutations, particularly MMR, underpin MSI and altered therapeutic responses [23,37,38,39,40,41,42].

Drug transport genes (*ABCB1*, *ABCC* family members, *ABCG2*, *SLC1A5*, *SLC29A1*, *SLC38A1*, *SLC39A4*, *SLC7A5*) were added as they encode influx and efflux transporters, with overexpression driving multidrug resistance [43]. Genes associated with epithelial–mesenchymal transition (EMT) and invasion (*CDH1*, *CDH2*, *ENO1*, *FLNA*, *SNAI2*, *ZEB1*, *ZEB2*) were included for their role in metastatic spread through loss of adhesion and enhanced motility [44,45]. Stress adaptation was addressed by incorporating endoplasmic reticulum (ER) stress response genes (*ATF4*, *ATF6*, *DDIT3*, *DNAJB9*, *DNAJC3*, *EIF2AK3*, *ERN1*, *HERPUD1*, *HSPA5*, *XBP1*), which regulate the unfolded protein response and survival under proteotoxic stress [46,47], and heat shock response genes (*DNAJA1*, *DNAJB1*, *HSF1*, *HSP90AA1*, *HSP90AB1*, *HSPA1A*, *HSPA1B*, *HSPB1*, *HSPB8*), which encode chaperones supporting protein homeostasis and therapy resistance [48]. Oxidative stress response genes (*GCLC*, *GCLM*, *GPX* family, *HMOX1*, *KEAP1*, *NFE2L2*, *NQO1*, *PRDX* family, *SOD1*, *SOD2*, *TXN*, *TXNRD1*) were selected for their central role in redox balance and tumor adaptation to ROS induced by chemotherapy [49].

In addition, metabolic regulators of folate and one-carbon metabolism (*ALDH1L1*, *ALDH1L2*, *BHMT*, *CBS*, *CHDH*, *CTH*, *DHFR*, *FOLH1*, *FOLR1*, *FPGS*, *GGH*, *MAT1A*, *MTHFR*, *MTR*, *MTRR*, *PEMT*, *PLD1*, *PLD2*, *RFC1*, *SHMT1*) were integrated to capture their influence on nucleotide biosynthesis, methylation, and 5-FU/folate therapy efficacy [50,51,52]. Similarly, genes involved in trans-sulfation and pyrimidine metabolism (*NFE2L2*, *RBP7*, *RRM2*, *SLC7A9*) were included for their role in regulating nucleotide pools and redox states [53,54,55,56]. Several signaling pathway genes (*APEX1*, *AXIN1*, *AXIN2*, *BRAF*, *CSNK1A1*, *CTNNB1*, *EGFR*, *GSK3B*, *MAP2K1*, *MAP2K2*, *MAPK14*, *MAPK3*, *PIK3CA*, *RAC3*, *RNF43*, *WNT3A*, *WNT5A*) were also selected as key nodes of *MAPK*, *PI3K*, and Wnt/β-catenin cascades commonly dysregulated in CRC [57]. Finally, core tumor suppressors (*APC*, *PTEN*, *RUNX3*, *SMAD4*, *TP53*) were incorporated for their central role as gatekeepers of proliferation and differentiation, whose dysregulation is related to chemoresistance in CRC [58,59,60,61]. We also included previously reported resistance-associated genes (*CCL22*, *CHGB*, *CSH2*, *FABP7*, *GALP*, *HSPA8*, *ICAM2*, *ICOS*, *LTBR*, *RABEP2*, *RARB*, *RELA*), as prior evidence has identified their role in conferring intrinsic or acquired resistance to 5-FU [62].

We extracted the gene expression data for these selected 180 genes (described above) from our previous studies [29,31,63], where we used the Illumina HT12 v4 BeadChip (Illumina Inc., San Diego, CA, USA) for the gene expression experiment. Tumor and normal samples from the same individuals were processed on the same chip. For cRNA synthesis, the Illumina^®^ TotalPrep RNA Amplification Kit (Ambion, a part of Life Technologies Corporation, Carlsbad, CA, USA) was used.

### 2.2. MSI Detection

For MSI detection, we used PCR followed by a high-resolution melt (HRM) analysis method. We used three MSI markers—*BAT25*, *BAT26*, and *CAT25*, as described in earlier studies [64]. *BAT25* and *BAT26* are the most widely used quasi-monomorphic mononucleotide repeats in the Bethesda panel for the identification of MSI [64]. *CAT25* was described by Findeisen et al. as displaying a quasi-monomorphic repeat pattern in normal tissue [64]. An earlier study confirmed the efficacy of the *CAT25* marker [65]. The thermocycling conditions included 95 °C for 2 min for enzyme activation, followed by 5 cycles of denaturation at 95 °C for 15 s, annealing starting at 60 °C for 30 s, extension at 72 °C for 30 s, and an additional 33 cycles of denaturation at 95 °C for 15 s, annealing at 53 °C for 30 s, and extension at 72 °C for 30 s. Before the HRM step, the products were heated to 95 °C for 1 min and cooled to 40 °C for 1 min, to allow heteroduplex formation. HRM was carried out, and the data were collected over the range from 60 °C to 95 °C, with a temperature increment of 0.2 °C/s at every 0.05 s [30,66]. A total of 18 tumor samples showed MSI for *BAT25* or *BAT26* markers, and all were confirmed by *CAT25*.

### 2.3. Statistical Analysis

For categorical variables, we used the chi-square test. For continuous variables, a *t*-test or one-way analysis of variance (ANOVA) was used. For gene expression data, we used the Partek Genomics Suite (v7.0) (https://www.partek.com/partek-genomics-suite/, accessed on 22 July 2025). Fold change (FC) with 95% confidence intervals (CIs) is reported. Different clinicopathological variables like MSI status, KRAS mutation status, TP53 mutation status, as well as clinical and pathological factors such as patient age (<40 years versus >40 years), tumor location (left- versus right-sided CRC), and disease stage were used as categorical variables in different ANOVA models. To determine if a given factor (e.g., MSI status) significantly influences the magnitude of differential expression between CRC and normal tissue, we introduced an interaction term Tissue × factor in the ANOVA model(s), where the *p*-value of the interaction term indicates if the difference between the magnitudes of differential expression of the gene significantly differs by the presence or absence of that given factor. For multiple testing correction, we used the FDR.

In the GO enrichment analysis, we tested whether a list of differentially expressed genes fell into a Gene Ontology category more often than expected by chance [67]. We used a chi-square test to compare the “number of significant genes from a given category/total number of significant genes” vs. “number of genes on chip in that category/total number of genes on the microarray chip”.

The gene set ANOVA is a mixed model ANOVA that compares expression levels of a “set of genes” instead of an individual gene in different groups (https://www.partek.com/partek-genomics-suite/, accessed on 22 July 2025). The result is expressed at the level of the “gene set” category by averaging the member genes’ results.

## 3. Results

The characteristics of the 71 CRC patients are summarized in Appendix A. Paired comparisons between CRC tissues and their corresponding non-lesional tissues were conducted to evaluate differential gene expression profiles. The results of this paired analysis of 380 probes covering all 180 genes, including fold changes and 95% confidence intervals, revealed that 105 probes were at least 1.2-fold differentially expressed in either direction at the FDR 0.05 level, and are presented in Appendix A. To account for multiple testing, false discovery rates (FDR) and adjusted *p*-values are also provided.

In the next step, instead of a single gene-level comparison, we asked if the 14 sets of genes (sharing similar biological pathways, e.g., apoptotic genes), on average, were differentially expressed in CRC tissue compared to corresponding normal tissue from the same patient (see Table 1). We used Gene set ANOVA for this analysis. We found that 11 out of the tested 14 sets of genes were dysregulated (upregulated = 8 and downregulated = 3) with a statistically significant *p*-value (<0.05) in either direction, with FDR 0.05 (see Table 1). Examples from among the upregulated gene sets are “Cell Cycle Checkpoint”, “Heat Shock Response”, “Oxidative Stress Response”, “Signaling pathway”, and examples from downregulated gene sets include “Tumor suppressor”, “Apoptotic gene”, “Endoplasmic Reticulum (ER) Stress”. It may be noted that 11 of these gene sets were also picked up by the enrichment analysis.

Considering the clinical significance of KRAS and TP53 mutations as well as MSI status in CRC, we examined whether these molecular features of the tumor affect the magnitudes of differential expression.

To assess whether KRAS mutation status modifies the extent of differential gene expression between CRC and non-lesional tissue, we incorporated an interaction term—“tissue type (1 = CRC, 0 = normal) × KRAS mutation status (1 = mutant, 0 = wild-type)”—into the ANOVA model. This allowed us to determine whether the presence of a *KRAS* mutation significantly altered the magnitude of differential expression. Our analysis identified three upregulated gene sets (see Table 2) for which the average differential expression between CRC and non-lesional tissues was significantly more pronounced in *KRAS*-mutated tumors (ANOVA interaction *p* < 0.05, Table 2).

Similarly, to examine the impact of *TP53* mutation status, we introduced a similar interaction term—“tissue type (1 = CRC, 0 = normal) × *TP53* mutation status (1 = mutant, 0 = wild-type)”—into the model. This allowed us to determine whether *TP53* mutations significantly alter gene expression patterns between CRC and normal tissues. We found that for three gene sets, the magnitude of differential expression was significantly greater in the presence of *TP53* mutations (ANOVA interaction *p* < 0.05, Table 3).

We further investigated the influence of another key molecular marker—MSI status—on the differential expression of different gene sets. We identified significant alterations in multiple gene sets when comparing CRC to normal tissues. Notably, gene sets related to Cell Cycle Checkpoint control and DNA Repair Mechanisms were slightly more upregulated in the presence of MSI, indicating an enhanced cellular response to genomic instability. Conversely, Oxidative Stress Response-related genes were more upregulated, and Tumor Suppressor genes were more markedly downregulated in MSS patients; see Table 4.

Finally, to assess the influence of broader clinicopathological factors—including age at diagnosis (early-onset < 40 years vs. late-onset ≥ 40 years), tumor grade (high vs. low), peri-neural invasion (present vs. absent), sex (male vs. female), presence of signet ring cells (present vs. absent), cancer stage (I, II, or III), telomere shortening (yes vs. no), and tumor infiltration (yes vs. no)—on gene expression differences between CRC and non-lesional tissue, we introduced an interaction term into the ANOVA model: tissue type (CRC = 1, normal = 0) × mutation status or presence/absence of the factor (1 = present, 0 = absent). This allowed us to assess whether these factors modify the extent of differential gene expression. Our analysis revealed that gene sets were differentially expressed depending on these variables; detailed results are presented in Appendix A.

### 3.1. Association of MSI Status on Differential Expression of Genes

Considering the clinical importance of MSI in CRC, we also examined the individual gene-level data (in addition to gene set level, described earlier), whether the expression of folate- and 5-FU-related genes differ between MSI and MSS tumors, based solely on gene expression profiles, without assessing actual chemotherapy response.

In folic acid metabolism-related genes, differential expressions vary significantly by the presence or absence of MSI (at FDR 0.05 level) and are shown in Table 5. The fold changes (FC) with 95% CI for each gene are presented. We also included the location (right-sided/left-sided tumor) into account. The GO Enrichment analysis (see Figure 1) showed that the list of genes was enriched in genes related to cell cycle genes (*BUB3*, *CDKN1A*, *SMAD4*); apoptosis (e.g., *DFFA*, *TNFRSF10B*, *ATF4*, etc.); Folate & One carbon metabolism (*TYMS*, *SHMT1*); p53 signaling pathway (*TNFRSF10B*, *CDKN1A*), etc.

Among the cell cycle-related genes, *BUB3* (see Figure 2A) was significantly upregulated only in MSI, whereas *SMAD4* was significantly downregulated only in MSS CRC (see Figure 2B). *PARP2* and *RRM2* were upregulated in both MSS and MSI CRCs, but the magnitude of upregulation was more marked in MSI tumors compared to MSS tumors (ANOVA interaction *p* < 0.001; see Figure 2C and Figure 2D, respectively).

In the apoptotic genes group, *FAS* was downregulated in both MSS and MSI tumors, with a stronger effect in MSS [FC = −1.58 (95% CI −1.74 to −1.44) in MSS patients compared to FC = −1.32 (95% CI −1.13 to −1.33) in MSI patients, ANOVA interaction *p* = 0.001]. *TNFRSF10B* was significantly upregulated in MSI (FC = 1.40, 95% CI 1.25 to 1.55, *p* = 6.09 × 10^−9^). *DFFA* showed subtype-specific regulation, which is upregulated in MSS (FC = 1.10, 95% CI 1.01 to 1.21, *p* = 0.07), but downregulated in MSI (FC = −1.22, 95% CI −1.45 to −1.03, *p* = 0.01).

Among the 5-FU metabolism genes, *TYMS* was significantly more upregulated in MSI CRC than in MSS tumors, (FC = 1.65, 95% CI 1.27–2.13 in patients with MSI tumors vs. FC 1.19 95% CI 1.02–1.39 in patients with MSS tumor; ANOVA interaction *p* = 1.01 × 10^−6^) even after adjustment for tumor location (see Figure 3A). Similarly, *PEMT* expression was higher in MSI tumors (FC = 1.23, 95% CI 1.11–1.38) than in MSS tumors (FC = 1.08, 95% CI 1.01–1.16; ANOVA interaction *p* = 0.002). In contrast, the one-carbon metabolism-related gene *SLC38A1* was significantly more downregulated in MSI tumors compared to MSS (MSI: FC = −1.59, 95% CI −1.91 to −1.33 vs. MSS: FC = −1.05, 95% CI −1.17 to −1.06; ANOVA interaction *p* = 2.2 × 10^−4^), independent of tumor location. *CDKN1A* was weakly upregulated in both MSI and MSS CRC, though the effect was more pronounced in MSI tumors (FC = 1.14, 95% CI 1.06–1.22; *p* < 0.001; see Figure 3B). Notably, *FOLR1* expression was specifically higher in MSS CRC compared to MSI tumors across both probes (ILMN_1661733 and ILMN_1698608) (Figure 3C,D).

In the stress response pathway, *ATF4* and *SOD1* were significantly upregulated in MSI compared to MSS CRCs. For *ATF4*, expression was elevated in MSI tumors (FC = 1.27, 95% CI 1.10–1.45) versus MSS (FC = 1.65, 95% CI 1.27–2.13; ANOVA interaction *p* = 1.43 × 10^−4^) (See Table 5). Similarly, *SOD1* expression was higher in MSI tumors (FC = 1.20, 95% CI 1.04–1.40) compared to MSS (FC = 1.04, 95% CI −1.05–1.14; ANOVA interaction *p* = 8.6 × 10^−4^). Notably, *MAPK3*, a key component of the *MAPK*/*ERK* pathway, was markedly downregulated in MSI tumors compared to MSS (MSI: FC = −2.02, 95% CI −2.46 to −1.67 vs. MSS: FC = −1.52, 95% CI −1.70 to −1.35; ANOVA interaction *p* = 3.75 × 10^−4^), with a statistically significant interaction. In contrast, *NQO1* showed divergent expression patterns, was downregulated in MSI tumors (FC = −1.38, 95% CI −1.92 to −1.67) but upregulated in MSS tumors (FC = 1.68, 95% CI 1.38–2.04; ANOVA interaction *p* = 2.22 × 10^−6^).

### 3.2. Association of MSI Status and KRAS Mutation Status on Differential Expression of Genes

Considering the clinical importance of MSI and *KRAS* mutation in CRC, we also looked for interactions between these two molecular markers. Based on the presence or absence of these two markers, the patients were divided into four groups: (a) MSS and *KRAS* wild (*n* = 38); (b) MSS and *KRAS* mutant (*n* = 15); (c) MSI and *KRAS* wild (*n* = 13) and (d) MSI and *KRAS* mutant (*n* = 5). Table 6 shows the magnitude of differential expressions (tumor vs. normal) of the genes that were significantly different among these four categories of CRC patients. From a 5-FU therapeutic point-of-view, the result suggests that (a) CRC patients with MSI, and more specifically MSI with *KRAS* mutation may benefit from therapy that can reduce *TYMS* or *BUB3* expression; and from therapy that can enhance *NQO1* expression; and (b) CRC patients with MSS irrespective of *KRAS* mutation status may benefit from therapy that may increase *SMAD4* expression.

### 3.3. Association of Age of Onset (<40 Years and >40 Years) on Differential Expression of Genes

At the individual gene level, the age of onset (EOCRC vs. LOCRC) significantly influenced (at *p* < 0.05) the magnitude of differential expression (CRC vs. normal tissue) of 29 out of 180 folic acid-related genes tested (see Appendix A). GO enrichment analysis of the pathways that involve this list of genes is presented in Appendix A. Among these genes, *MSH2* was upregulated only in LOCRC (FC = 1.10, 95% CI 1.05–1.15), but not in EOCRC (FC −1.00, 95% CI −1.06–1.06), ANOVA interaction *p* = 0.0001) (Figure 4). Notably, *PLD1*, *MAPK3*, *CHGB*, and *TNFSF10* were strongly downregulated in the older patients, whereas *BIRC5*, *ENO1*, and *CDK4* were markedly upregulated. In contrast, among early-onset CRC (<40 years), *GADD45A* was downregulated while *HSPA1A* was significantly upregulated (see Appendix A).

Enrichment analysis revealed that younger CRC patients (<40 years) exhibited significantly higher enrichment scores for folate biosynthesis (ES = 5.93), antifolate resistance (ES = 9.22), and colon cancer signaling (ES = 12.32) compared with older patients (>40 years) (Appendix A). This indicates the preferential activation of these pathways in the younger CRC patients.

## 4. Discussion

In CRC, 5-FU remains the backbone of chemotherapy in both adjuvant and palliative settings [68]. Genetic variations in genes regulating enzymes involved in 5-FU and folate metabolism play pivotal roles in shaping treatment response and clinical outcomes [69]. In this study, we tested the differential expression of a large number of genes related to folic acid that comprehensively cover different gene sets implicated in 5-FU resistance and folate metabolism. We also investigated whether their differential expressions vary by key clinical, pathological, and molecular genomic features, including age, sex, MSI status, *KRAS* mutation, *TP53* status, and telomere shortening in non-metastatic CRC. Our analysis revealed significant upregulation of cell cycle checkpoint genes, heat shock response genes, oxidative stress response genes, and signaling pathway genes, while tumor suppressor genes were consistently downregulated in CRC tissues compared with adjacent non-lesional tissues. Importantly, MSI tumors demonstrated distinct expression profiles compared with MSS tumors, particularly across pathways involving the p53 signaling pathway, cell cycle regulation, DNA damage response, apoptosis, stress signaling, and one-carbon metabolism. These findings reflect fundamental biological divergence between the two CRC subtypes and are supported by previous research and carry significant implications for the development of precise therapeutic strategies in CRC. Our major limitation is the lack of follow-up data, and, therefore, we could not test the effect of gene expression pattern of the tumor on disease outcome or therapeutic implications. Also, we did not perform any cell line study to address the mechanistic relationships. The study does not show causal relationships; rather, it shows some clinically relevant associations.

We acknowledge the fact that both (a) possible inhibition of antitumor effector responses (related to immune checkpoint inhibition) and (b) epigenetic factors are important aspects from a therapeutic point of view for CRC. But, in this study, we intentionally focused only on Folic acid and 5-FU-related genes and their differential expression. In fact, using the same patients, in a recent study, we have shown the interaction of MSI status and differential expression of Inflamed T-cell-related genes, which suggested that the MSS patients were less likely to benefit from immune checkpoint inhibitor (ICI) therapy compared to patients with MSI [63]. Similarly, in another previous study, we also described the interaction between MSI and epigenetic alteration (methylation) in these CRC patients, which suggested an opportunity for potential use of certain immune checkpoint inhibitors (CTLA4 and HAVCR2 inhibitors) in CRC with MSI [70].

### 4.1. Folate and One-Carbon Metabolism

In paired analyses, distinct expression patterns of 5-FU metabolism and folate cycle-related genes were observed between MSI and MSS tumors. Our data clearly showed more pronounced upregulation (CRC tissue compared to normal tissue) of *TYMS* in patients with MSI tumors compared to patients with MSS tumors, even after taking the location of the tumor into account. This partly explains the lesser effectiveness of 5-FU therapy in MSI patients. A previous study also suggested that thymidylate synthase (*TYMS*), the primary target of 5-FU [71], was more strongly upregulated in MSI tumors. *TYMS* overexpression is a well-recognized mechanism of 5-FU resistance, as it outcompetes fluorodeoxyuridylate (FdUMP) binding [72,73]. This may indicate that MSI tumors are relatively less sensitive to 5-FU due to an enhanced nucleotide biosynthesis capacity [74]. Another gene, phosphatidylethanolamine methyltransferase (*PEMT*), which regulates membrane phospholipid metabolism and methyl group flux [75], was also upregulated in MSI tumors. Emerging evidence suggests that *PEMT* dysregulation may alter membrane fluidity and influence drug uptake and efflux pathways, thereby potentially modulating chemosensitivity [76]. However, this requires further investigation. Conversely, solute carrier family 38 member 1 (*SLC38A1*), an amino acid transporter critical for glutamine uptake [77], was more downregulated in MSI tumors. Since glutamine availability fuels nucleotide biosynthesis and maintains redox balance, *SLC38A1* suppression could limit metabolic support for proliferation [78]. This may, paradoxically, sensitize MSI tumors to 5-FU under certain contexts, although compensatory pathways could offset this effect, warranting further exploration. Interestingly, serine hydroxy-methyltransferase 1 (*SHMT1*), which links serine/glycine metabolism to folate-mediated one-carbon metabolism [79], was downregulated in MSS tumors. Its reduced expression may disrupt one-carbon flux, potentially impairing DNA synthesis and repair [80]. This observation is consistent with prior clinical findings that MSS tumors often exhibit relatively higher 5-FU responsiveness compared with MSI tumors [81]. Overall, these gene expression differences may explain why MSI tumors are relatively resistant to 5-FU, whereas MSS tumors may remain more sensitive, emphasizing the need for further studies to guide subtype-specific therapies.

### 4.2. Cell Cycle Regulation and Checkpoint Control

Among the 180 genes analyzed, 38 were associated with cell cycle checkpoint regulation, and these were generally upregulated in CRC tissues compared with adjacent non-lesional tissues (see Table 1). When analyzed according to MSI and MSS status, adjusted for tumor location (right vs. left), *BUB3*, *CDKN1A* (p21), and *RRM2* were significantly more upregulated in MSI tumors than in MSS tumors (see Table 5). *CDKN1A*, a well-established transcriptional target of *p53*, is both necessary and sufficient for *p53*-mediated transcriptional repression [82] and has been implicated in 5-FU resistance [83]. In MSI CRC, *CDKN1A* upregulation suggests enhanced checkpoint fidelity and stronger p53-dependent growth arrest, which may allow tumor cells to repair DNA damage induced by 5-FU, potentially contributing to relative chemoresistance [84]. Our analysis also showed that CRCs harboring mutant *TP53* exhibited overall upregulation of signaling pathway-related genes (Table 3), likely reflecting activation of alternative compensatory mechanisms in the absence of intact p53. These findings support the idea that enhanced *p53-CDKN1A* signaling in MSI tumors may reduce 5-FU sensitivity, whereas MSS tumors with lower CDKN1A expression may be more susceptible to 5-FU-induced cytotoxicity, consistent with prior evidence linking altered *p53* pathway activity to chemoresistance in MSI CRC [85]. Importantly, the differential expression of checkpoint regulators between MSI and MSS tumors remained significant after adjusting for tumor site, highlighting MSI status as an independent determinant of *CDKN1A*-mediated 5FU sensitivity. In our analysis, we observed contrasting roles for *RRM2* and *BUB3* in MSI CRC. According to Zuo et al., drug-resistant tumor cells often exhibit amplification of the *RRM2* gene and its promoter, resulting in elevated transcriptional activity and increased DNA synthesis [86]. Consistently, we found *RRM2* overexpression in MSI tumors, suggesting that it may contribute to chemoresistance by enhancing replication potential and promoting survival under 5-FU-induced stress. In contrast, *BUB3*, a spindle assembly checkpoint (SAC) regulator transcriptionally controlled by *YY2*, enforces checkpoint fidelity. Hyperactivation of the *YY2*/*BUB3* axis delays mitosis, increases chromosomal instability beyond tolerable thresholds, and induces tumor cell death, thereby enhancing drug sensitivity [87]. These findings indicate that *RRM2* overexpression may drive resistance, whereas *BUB3* hyperactivation may sensitize tumors to therapy. The opposing functional effects of these two genes in MSI CRC warrant further investigation to determine which pathway predominates.

### 4.3. DNA Damage Response and Repair

The cytotoxic effect of 5-FU is mediated primarily by inhibiting *TYMS*, resulting in thymidine depletion, uracil misincorporation, DNA replication stress, and subsequent strand breaks [88]. *PARP2* plays a pivotal role in repairing 5-FU-induced single-strand breaks through the base excision repair (BER) pathway and subsequently engages the downstream HR/NHEJ repair mechanism [13]. In our cohort, *PARP2* was significantly overexpressed in MSI CRC, which may represent a compensatory response to increased DNA damage repair demands. Another important gene, *SMAD4*, a central mediator of the *TGF-β* pathway and regulator of 5-FU sensitivity, also showed subtype-specific expression patterns. In our analysis, *SMAD4* was significantly downregulated in MSS tumors, whereas MSI tumors demonstrated a mild, non-significant upregulation. Previous studies have shown that *SMAD4* knockout or downregulation promotes 5-FU resistance in CRC models [89], suggesting that *SMAD4* loss in MSS tumors may contribute to reduced drug sensitivity, consistent with experimental evidence [90]. Similarly, *PTEN*, a key tumor suppressor that regulates *PI3K*/*AKT* signaling, displayed distinct expression profiles across subtypes. Although PTEN mutations are more frequently reported in MSI CRC [91], we observed overall PTEN downregulation in both groups, with more pronounced loss in MSS tumors. This aligns with recent findings that *circPTEN* is downregulated in CRC [92]. PTEN is also a functional target of several microRNAs that mediate chemoresistance. For instance, miRNA-17-5p directly suppresses *PTEN* following chemotherapy, promoting multidrug resistance [93]. Likewise, miRNA-193-3p downregulation has been shown to restore *PTEN* expression, reduce proliferation, and enhance apoptosis, thereby reversing 5-FU resistance [94]. In another study, miRNA-141-3p overexpression suppressed *PTEN* in resistant CRC cells, whereas its inhibition restored *PTEN* and improved sensitivity to 5-FU and oxaliplatin [95]. These findings, together with our observations, suggest that targeting *PARP2* with *PARP* inhibitors, restoring *SMAD4*-mediated apoptotic pathways, or counteracting PTEN loss through inhibition of the *PI3K*/*AKT* pathway may represent promising strategies to enhance treatment efficacy in CRC.

### 4.4. Apoptosis Regulation

Among the apoptotic gene sets studied, *DFFA*, *TNFRSF10B* (*DR5*), and *FAS* exhibited differential expression patterns depending on MSI status in CRC. *DFFA*, an inhibitory subunit of the DNA fragmentation factor complex (DFF), was downregulated in MSI CRC. Under normal conditions, *DFFA* is cleaved by caspase-3 to release DFFB, which facilitates endonuclease-mediated DNA fragmentation—a terminal step of chemotherapy-induced apoptosis, including that triggered by 5-FU. Reduced *DFFA* expression may therefore impair apoptotic execution despite upstream caspase activation, potentially contributing to partial 5-FU resistance in MSI CRC. This finding aligns with reports that MSI tumors, while often associated with a favorable prognosis, can display relative chemoresistance to 5-FU-based therapies [96]. In contrast, MSI CRC also showed significant upregulation of DR5, consistent with preserved p53-dependent apoptotic signaling [97]. Elevated DR5 expression relative to adjacent non-lesional tissues suggests enhanced sensitivity to 5-FU-induced apoptosis, in line with previous evidence [98,99,100]. Conversely, *FAS (CD95*) was significantly downregulated in MSS CRC, reflecting impaired extrinsic apoptotic signaling in this subgroup. As a critical mediator of the *FAS*/*FASL* pathway, and a transcriptional target of *p53* [101], *FAS* loss has been linked to 5-FU resistance in colon carcinoma models [102]. The contrasting patterns of *DR5* upregulation in MSI, *FAS* suppression in MSS, and DFFA downregulation in MSI highlight a complex balance between apoptosis initiation and execution. While these alterations may offset each other at the bulk level, they remain therapeutically relevant, suggesting that enhancing DR5 in MSI, restoring FAS in MSS, or overcoming DFFA defects could help tailor and improve 5-FU-based treatment responses.

### 4.5. Stress Response Pathways

In the studied samples, oxidative stress-related genes were generally upregulated in both MSI and MSS CRC, though more prominently in MSS tumors (Table 4). In MSI tumors, *ATF4* and *SOD1* were consistently upregulated, whereas *MAPK3* was strongly downregulated. Interestingly, *NQO1* displayed a biphasic pattern, being downregulated in MSI but upregulated in MSS. These findings indicate that oxidative stress responses differ according to MSI status. The oxidative stress pathway is increasingly recognized as a key determinant of chemotherapeutic sensitivity [103], and 5-FU is known to induce reactive oxygen species (ROS) that drive apoptosis in cancer cells [104]. Upregulation of *ATF4*, a component of the *PERK–ATF4* pathway linked to 5-FU resistance [105], along with *SOD1* overexpression, may reflect adaptive stress responses in MSI CRC, although *SOD1* is also frequently upregulated in early-stage CRC [106]. In contrast, *MAPK3* downregulation in MSI tumors may shift the balance between apoptosis and autophagy toward resistance [7]. Notably, MSS tumors exhibited stronger upregulation of *NQO1*, which likely functions as a compensatory antioxidant mechanism to counteract chemotherapy-induced ROS, thereby limiting apoptosis and conferring relative resistance [107,108]. Taken together, the combination of *MAPK3* downregulation and reduced *NQO1* expression in MSI tumors may impair ROS neutralization, resulting in greater oxidative damage and enhanced 5-FU sensitivity. Conversely, MSS tumors, through more robust antioxidant defenses, may mitigate ROS-mediated cytotoxicity and thus exhibit relative drug resistance. These results suggest that differential oxidative stress responses between MSI and MSS CRC contribute to their divergent therapeutic outcomes. Importantly, targeting oxidative stress regulators—such as inhibiting *NQO1* or *ATF4* in MSS tumors, or modulating *MAPK3* and ROS balance in MSI tumors—may provide opportunities for tailored therapeutic strategies to enhance 5-FU efficacy in a subtype-specific manner.

## 5. Conclusions

In conclusion, MSI colorectal tumors exhibited *TYMS* and *PARP2* upregulation and *DFFA* downregulation. These alterations may be linked to reduced 5-FU sensitivity through enhanced nucleotide biosynthesis, impaired apoptotic execution, and increased DNA repair activity. *FAS* expression was decreased in both MSI and MSS tumors, though the downregulation was more pronounced in MSS, suggesting a greater impairment of extrinsic apoptotic signaling in this subgroup. Additionally, MSS tumors show downregulation of *SMAD4*, which is partially offset by compensatory upregulation of *NQO1* and *GSK3B*. This distinct expression may explain subgroup differences in treatment response and suggest potential avenues for tailored therapy, such as *TYMS* or *PARP* inhibition in MSI tumors and restoration of apoptotic signaling or targeting compensatory pathways in MSS tumors.

## Figures and Tables

**Figure 1 curroncol-32-00661-f001:**
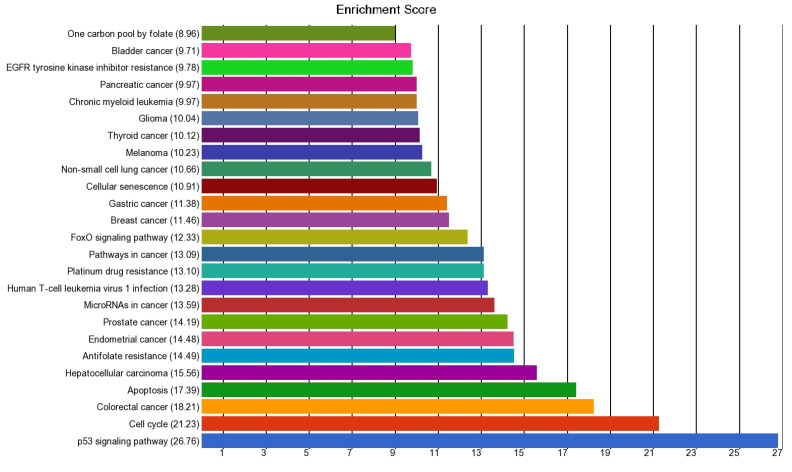
Enrichment Scores for pathways in patients with MSI.

**Figure 2 curroncol-32-00661-f002:**
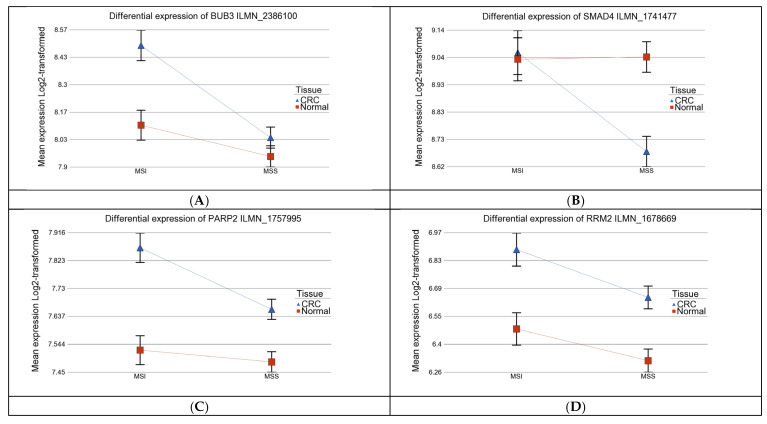
Differential gene expression in CRC versus adjacent normal tissue, stratified by MSI status and adjusted for tumor location. Tumor tissues are shown in blue and normal tissues in red. Expression profiles are shown for BUB3 (ILMN_2386100) (**A**); SMAD4 (ILMN_1741477) (**B**); PARP2 (ILMN_1757995) (**C**); and RRM2 (ILMN_1678669) (**D**).

**Figure 3 curroncol-32-00661-f003:**
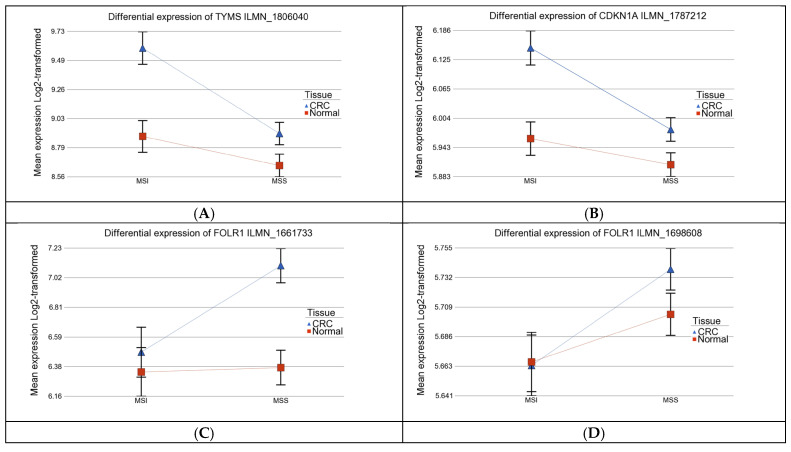
Differential gene expression in CRC versus adjacent normal tissue, stratified by MSI status and adjusted for tumor location. Tumor tissues are shown in blue and normal tissues in red. Expression profiles are shown for TYMS (ILMN_1806040) (**A**), CDKN1A (ILMN_1787212) (**B**), and FOLR1 (ILMN_1661733 and 1698608) (**C**,**D**).

**Figure 4 curroncol-32-00661-f004:**
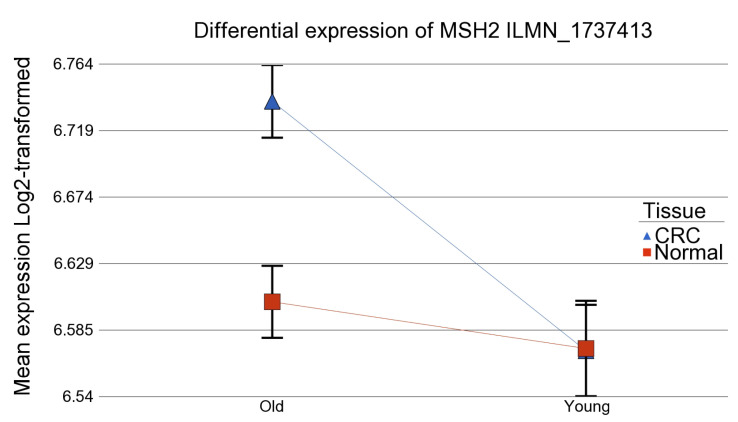
Differential expression for MSH2 ILMN_1737413 in patients < 40 years of age and >40 years of age.

**Table 1 curroncol-32-00661-t001:** Differential expression of gene sets between CRC tissue and adjacent non-lesional tissues.

GO Description	*p*-Value	FDR (*p*-Value)	Fold-Change	(95% CI)
Cell Cycle Checkpoint Genes	8.94 × 10^−217^	1.25 × 10^−215^	1.27	(1.25–1.29)
Heat Shock Response Genes	3.04 × 10^−53^	2.13 × 10^−52^	1.32	(1.27–1.37)
Oxidative Stress Response Genes	5.43 × 10^−33^	2.54 × 10^−32^	1.11	(1.09–1.13)
Signaling pathway	4.09 × 10^−23^	1.43 × 10^−22^	1.08	(1.06–1.10)
Tumor suppressor gene	5.08 × 10^−21^	1.42 × 10^−20^	−1.16	(−1.20–−1.13)
DNA Repair Mechanisms	1.17 × 10^−15^	2.73 × 10^−15^	1.04	(1.03–1.05)
Drug Transport	3.46 × 10^−10^	6.91 × 10^−10^	1.07	(1.05–1.10)
Folate & One carbon metabolism	6.65 × 10^−8^	1.16 × 10^−7^	1.04	(1.02–1.05)
Apoptotic gene	1.78 × 10^−6^	2.77 × 10^−6^	−1.02	(−1.03–−1.01)
Endoplasmic Reticulum (ER) Stress	0.010382	0.014534	−1.03	(−1.06–−1.01)
5FU Metabolism	0.016943	0.021564	1.04	(1.01–1.08)
Transsulfuration & Pyrimidine metabolism	0.05517	0.064365	1.06	(−1.00–1.13)
known 5FU-resistance genes	0.363536	0.3915	1.01	(−1.01–1.03)
EMT & tumor invasion	0.578479	0.578479	1.01	(−1.03–1.05)

**Table 2 curroncol-32-00661-t002:** Differential gene sets expression between *KRAS* Mutant and *KRAS* Wild CRC.

Gene Set	Interaction *p*-Value	FDR *p*-Value	*KRAS* Mutant		*KRAS* Wild	
			Fold Change	(95% CI)	Fold Change	(95% CI)
Cell Cycle Checkpoint Genes	1.09 × 10^−8^	1.53 × 10^−7^	1.33	(1.30 to 1.37)	1.24	(1.22 to 1.26)
Heat Shock Response Genes	4.57 × 10^−4^	3.20 × 10^−3^	1.45	(1.36 to 1.55)	1.26	(1.21 to 1.31)
Oxidative Stress Response Genes	2.65 × 10^−2^	1.24 × 10^−1^	1.14	(1.11 to 1.18)	1.09	(1.07 to 1.11)

**Table 3 curroncol-32-00661-t003:** Differential gene sets expression between *TP53* Mutant and TP53 Wild CRC.

Gene Set	Interaction *p*-Value	FDR *p*-Value	*TP53* Mutant		*TP53* Wild	
			Fold Change	(95% CI)	Fold Change	(95% CI)
Oxidative Stress Response Genes	1.69 × 10^−11^	2.36 × 10^−10^	1.21	(1.17 to 1.25)	1.05	(1.03 to 1.08)
Endoplasmic Reticulum (ER) Stress	1.28 × 10^−5^	8.98 × 10^−5^	1.05	(1.00 to 1.10)	−1.07	(−1.11 to −1.04)
Signaling pathway	2.61 × 10^−3^	1.22 × 10^−2^	1.12	(1.09 to 1.15)	1.06	(1.04 to 1.08)

**Table 4 curroncol-32-00661-t004:** Differential gene sets expression between MSI and MSS CRC.

Gene Set	Interaction *p*-Value	FDR *p*-Value	CRC with MSS		CRC with MSI	
			Fold Change	(95% CI)	Fold Change	(95% CI)
Cell Cycle Checkpoint Genes	3.72 × 10^−15^	5.21 × 10^−14^	1.26	(1.24 to 1.29)	1.27	(1.23 to 1.30)
Oxidative Stress Response Genes	5.30 × 10^−10^	3.71 × 10^−9^	1.11	(1.09 to 1.13)	1.08	(1.05 to 1.12)
Tumor suppressor gene	1.37 × 10^−4^	3.82 × 10^−4^	−1.19	(−1.24 to −1.15)	−1.08	(−1.14 to −1.02)
DNA Repair Mechanisms	4.96 × 10^−2^	7.71 × 10^−2^	1.04	(1.03 to 1.05)	1.05	(1.03 to 1.07)

**Table 5 curroncol-32-00661-t005:** Differential expression of folic acid metabolism-related genes between MSI and MSS in CRC.

Probeset ID	Gene	Interaction *p*-Value	FDR (*p*-Value)	CRC MSI vs. Normal MSI			CRC MSS vs. Normal MSS)		
				Fold-Change	(95% CI)	*p*-Value	Fold-Change	(95% CI)	*p*-Value
ILMN_2386100	BUB3	3.45 × 10^−7^	0.0001	1.31	(1.14–1.50)	0.000233	1.06	(−1.02–1.16)	0.13
ILMN_1806040	TYMS	1.01 × 10^−6^	0.0001	1.65	(1.27–2.13)	0.000181	1.19	(1.02–1.39)	0.02
ILMN_1720282	NQO1	2.22 × 10^−6^	0.0002	−1.38	(−1.92–1.01)	0.05	1.68	(1.38–2.04)	6.20 × 10^−7^
ILMN_2385220	DFFA	6.92 × 10^−6^	0.0005	−1.22	(−1.45–−1.03)	0.01	1.1	(−1.01–1.21)	0.07
ILMN_1699265	TNFRSF10B	1.67 × 10^−5^	0.001	1.69	(1.42–2.02)	2.17 × 10^−8^	1.17	(1.06–1.30)	0.002
ILMN_2331010	TNFRSF10B	5.03 × 10^−5^	0.002	1.4	(1.25–1.55)	6.09 × 10^−9^	1.16	(1.09–1.24)	5.29 × 10^−7^
ILMN_1741477	SMAD4	8.71 × 10^−5^	0.004	1.02	(−1.15–1.20)	0.8	−1.28	(−1.41–−1.16)	1.22 × 10^−6^
ILMN_2358457	ATF4	0.000143	0.005	1.27	(1.10–1.45)	0.00097	1.23	(1.13–1.34)	4.13 × 10^−6^
ILMN_1769911	SLC38A1	0.000202	0.01	−1.59	(−1.91–−1.33)	1.25 × 10^−6^	−1.05	(−1.17–1.06)	0.37
ILMN_1757995	PARP2	0.000298	0.01	1.27	(1.15–1.39)	1.68 × 10^−6^	1.13	(1.07–1.19)	3.50 × 10^−5^
ILMN_2402341	MAPK3	0.000375	0.01	−2.02	(−2.46–−1.67)	3.22 × 10^−11^	−1.52	(−1.70–−1.35)	4.32 × 10^−11^
ILMN_1787212	CDKN1A	0.000497	0.01	1.14	(1.06–1.22)	0.000216	1.05	(1.01–1.09)	0.01
ILMN_1667260	MAPK3	0.000665	0.01	−2.05	(−2.51–−1.67)	1.17 × 10^−10^	−1.55	(−1.75–−1.37)	4.89 × 10^−11^
ILMN_1743784	SHMT1	0.000782	0.01	1.02	(−1.07–1.12)	0.64	1.001	(−1.05–1.06)	0.97
ILMN_1662438	SOD1	0.00086	0.01	1.2	(1.04–1.40)	0.01	1.04	(−1.05–1.14)	0.36
ILMN_1701134	PTEN	0.001077	0.02	−1.2	(−1.33–−1.07)	0.001	−1.22	(−1.31–−1.15)	7.77 × 10^−9^
ILMN_2319077	FAS	0.001165	0.02	−1.32	(−1.56–−1.13)	0.0007	−1.58	(−1.74–−1.44)	1.64 × 10^−16^
ILMN_1811933	SHMT1	0.001383	0.02	−1.07	(−1.33–1.15)	0.51	−1.13	(−1.28–1.01)	0.06
ILMN_1779376	GSK3B	0.001934	0.03	1.03	(−1.10–1.18)	0.61	1.19	(1.10–1.28)	2.04 × 10^−5^
ILMN_1727855	PEMT	0.002	0.042	1.23	(1.11–1.38)	0.0002	1.08	(1.01–1.16)	0.015
ILMN_1678669	RRM2	0.002	0.042	1.32	(1.13–1.55)	0.0007	1.25	(1.13–1.37)	1.03 × 10^−5^

**Table 6 curroncol-32-00661-t006:** Differential expression of genes in relation to *KRAS* mutation status and microsatellite instability status in CRC vs. normal tissue.

Gene Symbol	Interaction *p*-Value	MSS in CRC vs. Normal		MSI in CRC vs. Normal	
		KRAS Wild	KRAS Mutant	KRAS Wild	KRAS Mutant
		FC (95% CI)	FC (95% CI)	FC (95% CI)	FC (95% CI)
*BUB3*	1.10 × 10^−5^	1.05 (−1.06 to 1.16)	1.11 (−1.05 to 1.30)	1.27 (1.08 to 1.49)	1.49 (1.12 to 1.99)
*NQO1*	2.24 × 10^−5^	1.51 (1.20 to 1.91)	2.16 (1.51 to 3.09)	−1.53 (−2.24 to −1.05)	−1.08 (−2.10 to 1.81)
*TYMS*	2.94 × 10^−5^	1.17 (−1.03 to 1.40)	1.27 (−1.04 to 1.68)	1.58 (1.17 to 2.12)	2.02 (1.19 to 3.42)
*TNFRSF10B*	2.00 × 10^−4^	1.11 (−1.02 to 1.26)	1.35 (1.11 to 1.63)	1.65 (1.35 to 2.02)	1.86 (1.30 to 2.66)
*DFFA*	3.00 × 10^−4^	1.10 (−1.03 to 1.24)	1.10 (−1.10 to 1.32)	−1.20 (−1.46 to 1.02)	−1.36 (−1.93 to 1.04)
*TNFRSF10B*	5.00 × 10^−4^	1.14 (1.06 to 1.23)	1.23 (1.09 to 1.38)	1.40 (1.24 to 1.59)	1.39 (1.12 to 1.73)
*GSK3B*	7.00 × 10^−4^	1.12 (1.02 to 1.23)	1.38 (1.20 to 1.59)	1.11 (−1.05 to 1.28)	−1.22 (−1.58 to 1.07)
*SMAD4*	1.10 × 10^−3^	−1.25 (−1.40 to −1.11)	−1.36 (−1.62 to −1.14)	−1.05 (−1.27 to 1.15)	1.29 (−1.08 to 1.80)

## Data Availability

All the supporting data are presented in the tables presented in the main manuscript and Appendix A.

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
