# Peer review of "Mutation and Microsatellite Instability (MSI) Affect the Differential Gene Expression of Folic Acid and 5-Flourouracil Metabolism-Related Genes in Colorectal Carcinoma"

_curroncol, 2025, doi:10.3390/curroncol32120661_

Round 1
Reviewer 1 Report
Comments and Suggestions for Authors
The article presents a good perspective and has reliable results, but it has gaps regarding effective factors such as possible inhibition of antitumor effector responses and and epigenetic factors, considering the age of the selected samples Needs clarification on this for better clarity.
Author Response
Thank you very much for the helpful comment and suggestions. Please see the attached file for our response.

Reviewer 2 Report
Comments and Suggestions for Authors
This manuscript presents a transcriptomic analysis of folic acid and 5-FU metabolism–related genes in colorectal carcinoma, stratified by microsatellite instability, KRAS, and TP53 mutation status. The study addresses a relevant and timely topic in precision oncology and provides novel insights into molecular differences that may influence 5-FU response across CRC subtypes. The work is generally well structured, and the methodology is detailed. However, there are several issues that should be addressed before acceptance.
- The manuscript adds to existing literature on 5-FU resistance by integrating MSI and mutational status with gene expression analysis. However, the novelty is somewhat limited, as many of the associations (e.g., TYMS upregulation in MSI, SMAD4 and PTEN downregulation) have been previously reported.
- The authors conclude that gene expression differences “may explain” differential 5-FU responses, but no treatment or survival data are provided.
- The manuscript describes using both ANOVA and Gene Set ANOVA. However, the rationale for using a 1.2-fold cutoff and FDR<0.05 should be justified.
Author Response
Thank you very much for the review comments and suggestions. Please see the attached file for our response.

Reviewer 3 Report
Comments and Suggestions for Authors
The authors of this manuscript describe an analysis of differential mRNA expression for genes related to 5-FU and folic acid metabolism in colorectal cancer, comparing tumor and matched normal tissue across key clinical and molecular subgroups. This manuscript provides an interesting overview of gene expression patterns in colorectal cancer and, in my opinion, could be suitable for publication after the authors address the minor and major issues outlined below:
- In Abstract there are many abbreviations that appear without prior clarification. Please make sure that all the abbreviations are clarified first time they appear.
- In materials and methods section the authors state that the details of sample collection and preservation are described in previous studies. This is fine but they still should describe it here since this is a new publication. Also please describe inclusion and exclusion criteria for this study.
- In the section 2.1. Gene Selection and Functional Relevance, there are lots of gene abbreviations that need to be clarified.
- In the section 2.2. MSI Detection the authors state:”A high-resolution melt (HRM) analysis method was used for the detection of MSI markers—BAT25, BAT26and CAT25, as described in earlier studies”. Please also describe it here since this is different publication and not just the continuation of the previous one.
- In statistical analysis section the authors state: “We used ANOVA, andGene set ANOVA as described in previous papers (29,30).” Please describe it briefly here.
- There are two tables named table 1. Please correct.
- Please correct the table references in the text, as they do not correspond to the results described in the paragraph (table 2, 3 etc.).
- Figures 1, 2 and 3 need to be improved to enhance legibility.
- Why did the authors mark SLC38A1 gene in bold letters in line 315?
- In the discussion section the authors make many strong statements. Given that their findings are based on observational mRNA expression data, causal relationships cannot be established, and any functional interpretations should be made with caution. Differences in transcript levels reflect associations rather than direct mechanistic effects, and should not be overinterpreted as evidence of causation. Please rewrite accordingly.
Author Response
Thank you very much for the helpful comments and suggestions. Please see the attached file for our responses.

Round 2
Reviewer 1 Report
Comments and Suggestions for Authors
The author's answer was convincing to me.
Reviewer 2 Report
Comments and Suggestions for Authors
I don't have further comments at the current stage.